# Ammonium Tetrakis(pentafluorophenyl)borate: Preparation and Application in Olefin Coordination Polymerization as the Cocatalyst Compound

**DOI:** 10.3390/polym16121689

**Published:** 2024-06-13

**Authors:** Yiming Wei, Shuzhang Qu, Xinwei Li, Jian Chen, Zhao Wen, Qian Li, Wei Wang

**Affiliations:** SINOPEC (Beijing) Research Institute of Chemical Industry Co., Ltd., No. 14 Beisanhuan Donglu, Chao Yang District, Beijing 100013, China; weiym.bjhy@sinopec.com (Y.W.); qushzh.bjhy@sinopec.com (S.Q.); lixw.bjhy@sinopec.com (X.L.); chenjian.bjhy@sinopec.com (J.C.); wenzh.bjhy@sinopec.com (Z.W.); liq.bjhy@sinopec.com (Q.L.)

**Keywords:** ammonium tetrakis(pentafluorophenyl)borate, metallocene catalyst, cocatalyst, olefin polymerization

## Abstract

Metallocene catalysts have attracted much attention from academia and industry for their excellent catalytic activity in the field of olefin polymerization. Cocatalysts play a key role in metallocene catalytic systems, which can not only affect the overall catalytic activity, but also have an obvious influence on the structure and properties of the polymer. Although methylaluminoxane (MAO) is currently the most widely used cocatalyst, its price increases the production cost of polyolefin materials. Ammonium tetrakis(pentafluorophenyl)borate has shown excellent performance in polymerization, being one of the best substitutes for the traditional cocatalyst MAO. Compared with the main catalyst, whose composition and structure are relatively complex, the research on cocatalyst is very limited. This review mainly introduces the research history, preparation methods, and application progress in polymerization of ammonium tetrakis(pentafluorophenyl)borate, deepening our understanding of the role of cocatalyst in polymerization, with the hope of inspiring brand-new thinking on improving and enhancing the overall performance of catalyst systems.

## 1. Introduction

Metallocene catalyst [1,2,3,4] usually refers to a catalytic system composed of metallocene compounds as the main catalyst and a Lewis acid as a cocatalyst, which was industrialized in the early 1990s of the 20th century [5]. The catalytic polymerization mechanism is that metallocene and cocatalyst interact to form the uniform cationic catalytic active site, namely the single active site, whose chemical environment is easy to regulate by adjusting the structures of ligands [6]. The polyolefin produced by single-site catalyst is a kind of polymer with incomparable advantages, such as various stereostructure regularities, few impurities, and uniform distribution of molecular weight and chemical composition [5,6,7].

In addition to the main catalyst, the cocatalyst also plays an indispensable role in the metallocene catalytic system. Numerous research results indicate that the activation effect of cocatalysts mainly manifests in the following three steps: (1) the alkylation of cocatalysts on metal centers; (2) the cocatalyst’s extraction of alkylation groups from metal centers; (3) the formation by the cocatalyst part of a pair of anions and cations with the metal center part [8].

Methylaluminoxane (MAO) was first applied as an efficient cocatalyst for metallocene olefin polymerization [9]. In the late 1970s, alkylaluminum and metallocene compounds were used for olefin polymerization at the Universität Hamburg in Germany [8]. Later, it was determined that the activation effect was obtained from the hydrolysis of trimethylaluminum [10]. From then on, MAO-activated metallocene catalysts have been widely used in the field of olefin polymerization [11]. The hydrolysis of trimethylaluminum to obtain MAO also inspired the synthesis of different aluminoxanes from other alkyl aluminum, such as ethyl aluminoxane (EAO) [12], or the co-hydrolysis of a variety of alkylaluminums to obtain modified MAO (MMAO) [13]. Further processing with MAO can also yield different alkylaluminoxanes, such as MAO with volatile components removed (DMAO) [14]. These alkylaluminoxanes can exhibit different catalytic performance due to their unique properties and structures [13,14,15,16,17,18,19,20]. However, their application is too specific to apply on a large scale [8], and the activation effect of most cocatalysts is significantly lower than that of MAO. Therefore, MAO remains the most widely used cocatalyst in the field of olefin polymerization. But these MAO cocatalysts are expensive and also come with certain safety risks. In industrial production, the cocatalyst MAO significantly increases the cost of polyolefin materials. Therefore, it is of vital importance to develop a new type of economical cocatalysts with better active performance for replacing MAO.

Perfluoroaryl borates are a new generation of cocatalysts and regarded as the ideal alternative to MAO [21]. As early as the early 1990s, it was discovered that such cocatalysts may be active for olefin polymerization [22]. At first, it turned out that [B(C_6_H_5_)_4_]^−^ was used as a cocatalyst to catalyze the polymerization of propylene, but the catalytic efficiency was not ideal [23]. Afterwards, the perfluorinated tetraphenylborate as an anion had made a breakthrough in this area and was reported to have high catalytic activity [24]. At present, the most frequently used borate as a cocatalyst is tetrakis(pentafluorophenyl)borate, whose anion is [B(C_6_F_5_)_4_]^−^, and the cations are mainly inorganic ion, carbonium ion, and organic ammonium ion. Commonly used products include sodium/potassium tetrakis(pentafluorophenyl)borate (Na^+^/K^+^ [B(C_6_F_5_)_4_]^−^), *N*,*N*-dimethylphenylammonium tetrakis(pentafluorophenyl)borate (PhNHMe_2_^+^ [B(C_6_F_5_)_4_]^−^), triphenylcarbonium tetrakis(pentafluorophenyl)borate (Ph_3_C^+^ [B(C_6_F_5_)_4_]^−^), and so on [25]. Compared to MAO, these borate systems also exhibit remarkable catalytic activity. On the one hand, the strong electron-withdrawing effect of the fluorine on the perfluorobenzene ring reduces the electron cloud density of the boron atom, thus greatly weakening the “complex force” with the metal ion. Additionally, perfluorophenyl group has a large steric hindrance, further inhibiting the “complex force” between each other, which is more conducive to the formation of the metal center of the cation and the ion pair of the anion [6]. Under the synergism of electron effect and steric effect, the electrophilicity of the active site of the metal is enhanced, which is more beneficial to the coordination with the double bond of the olefin monomer to the active site, thereby improving the catalytic performance. 

The activation mechanism of MAO and perfluoroaryl borate as a cocatalyst is illustrated in Figure 1. The transition metal complex was firstly methylated by MAO to generate L_2_MMeCl, and then Cl^−^ was abstracted by Al in MAO to form the transition metal cation, which could react with the monomer, realizing the chain initiation and propagation, as presented in Figure 1a. When ammonium perfluoroaryl borate was used as the cocatalyst, the transition metal complex would be firstly alkylation by alkyl aluminum. Then, the transition metal cation could be obtained through the dealkylation by ammonium cation, which could form a looser ion pair with the borate and meanwhile generate equimolar amine and alkane, as presented in Figure 1b [26].

This review mainly describes the preparation methods and research progress of perfluoroaryl borate, especially ammonium tetrakis(pentafluorophenyl)borate as cocatalyst, and analyzes and prospects the future development of cocatalyst in the field of olefin polymerization. In comparison with previous reports [6,8,26], the differences between MAO and perfluoroaryl borate were fully compared, and the research progress of perfluoroaryl borate as a cocatalyst to replace MAO in polymerization is introduced in detail in this paper. It also points out the research areas where perfluoroaryl borate has not been fully applied.

## 2. Research History of Ammonium Tetrakis(pentafluorophenyl)borate

In 1986, Echols and Scott et al. [27,28] prepared Cp_2_Zr(Me)(MeCN)_2_**^+^**BPh_4_^−^ by reacting Ag^+^BPh_4_^−^ with Cp_2_ZrMe_2_, and reported the metallocene cation active site stabilized by borate for the first time. However, the Lewis acidity of BPh_4_^−^ is weak, with obvious electron-donating effect, and the interaction is strong with Cp_2_ZrMe^+^, which cannot form a loose ion pair structure. Therefore, Cp_2_Zr(Me)(MeCN)_2_^+^BPh_4_^−^ has low catalytic activity for olefin polymerization. In order to reduce the interaction between the cationic active center and the coordination anion and form a looser ion pair structure to improve the catalytic activity, a large number of fluorinated boranes or borates were designed and synthesized. Several typical perfluoroarylboranes and borates are illustrated in Figure 2. 

B1 was one of the first perfluoroboranes to be utilized as the cocatalyst to activate metallocene catalysts. Although its synthesis was reported earlier [29,30,31], it was not until 1991 that B1 was used to activate metallocene catalysts for olefin polymerization [22,24], showing high catalytic activity. In order to improve the Lewis acidity of B1 and increase its steric hindrance, Marks et al. further introduced perfluoroaryl into B1, designing and synthesizing B2 [32], B3 [21], and B4 [33]. Compared with B1, the Lewis acidity and steric hindrance of B2–B4 are obviously increased. Therefore, their catalytic activity could be obviously improved. 

Later, several diboranes such as B5 [34] and B6 [35] were designed and synthesized, Because the interaction between adjacent perfluorophenyl groups inhibited the conjugate electron-donating effect of the perfluorophenyl group to boron atoms, the Lewis acidity of B5 and B6 was further increased, and the catalytic activity of polymerization with metallocene was also significantly increased. 

B1 interacts with Cp_2_ZrMe_2_ to form the complex Cp_2_ZrMe^+^[MeB(C_6_F_5_)_3_]^−^. Due to the strong electron-donating effect of the methyl group in [MeB(C_6_F_5_)_3_]^−^, the interaction between [MeB(C_6_F_5_)_3_]^−^ and the cationic active center is still obvious. In order to further decrease the electron-donating effect of the coordination anions, Rausch and Marks et al. designed and synthesized perfluorophenylborate, such as B7 [24,36]. B7 interacts with Et(Ind)_2_ZrMe_2_ to form Et(Ind)_2_ZrMe^+^B(C_6_F_5_)_4_^−^. Due to the poor electron-donating effect of B(C_6_F_5_)_4_^−^, it can form a very loose ion pair structure with the cationic active center Et(Ind)_2_ZrMe^+^. Therefore, the catalytic activity of borates is always higher than that of borane under the same experimental conditions. However, the solubility and thermal stability of B7 in aliphatic solvents still needs further improvement. In order to compensate for these two deficiencies, B8 [37] was designed and synthesized. Compared with B7, the solubility and thermal stability of B8 in aliphatic solvent were significantly improved; the catalytic activity of B8 cannot exceed that of B7, and it has to be used under certain conditions. At present, B7 is still a widely used cocatalyst in the olefin polymerization field.

## 3. Preparation of Ammonium Tetrakis(pentafluorophenyl)borate

Generally speaking, the usual method of preparing ammonium tetrakis(pentafluorophenyl)borate is to first prepare the tetrakis(pentafluorophenyl)borate anion and the organic cation and then exchange the ions to obtain the final product. This chapter mainly focuses on the preparation of the dimethylanilinium tetrakis(pentafluorophenyl)borate (PhNHMe_2_^+^[B(C_6_F_5_)_4_]^−^, DMAB, see Figure 3) and *N*-methyl-*N*,*N*-dioctadecylammonium tetrakis(perfluorophenyl)borate ([Me(C_18_H_37_)_2_NH]^+^[B(C_6_F_5_)_4_]^−^, MDOAB, see Figure 4), respectively, from perspective of anions and cations.

### 3.1. Preparation of Tetrakis(pentafluorophenyl)borate Anions

The preparation of the tetrakis(pentafluorophenyl)borate anion is the crucial step in the synthesis of the final product. Pentafluorobenzene and pentafluorohalobenzene are the common reactants, the dehydrogenation or dehalogenation of which is necessary. The preparation methods are as follows.

#### 3.1.1. Pentafluorobenzene Dehydrogenation

Butyl lithium reagent with strong reactivity is the contributor to the dehydrogenation of pentafluorobenzene, and at low temperature, lithium pentafluorophenyl intermediate (LiC_6_F_5_) can be formed with *t*-butyl lithium reagent. Tosoh Akzo Corp [38] reported a method for preparing the tetrakis(pentafluorophenyl)borate anions. The LiC_6_F_5_ intermediate is prepared by dehydrogenation in *n*-hexane at −55 °C; subsequently, the borate anion is obtained by boronization with boron trichloride. Similarly, Asahi Glass Co., Ltd. (Tokyo, Japan) [39] suggested a method for dehydrogenation with an increase in reaction temperature to −40 °C, which also obtained the target product.

However, LiC_6_F_5_ is an unavoidable key intermediate for this approach. It is worth noting that the LiC_6_F_5_ is potentially hazardous, and can decompose explosively if the temperature of the *n*-hexane or *n*-pentane solution is raised above −50 °C. It should be handled only on a small scale with appropriate safety precautions (safety shields, safety glasses, face shields, leather gloves, and protective clothing). Rapid changes in temperature can result in violent explosions, which is undoubtedly an impediment to industrial production.

#### 3.1.2. Pentafluorohalobenzene Dehalogenation

Besides pentafluorobenzene dehydrogenation, pentafluorohalobenzene dehalogenation is also a popular method for preparing ammonium tetrakis(pentafluorophenyl)borate. Butyl lithium and Mg can be common dehalogenation reagents. The dehalogenation methods for preparing tetrakis(pentafluorophenyl)borate anions are summarized in Table 1.

As indicated in Table 1, for dehalogenation of the pentafluorohalobenzene, pentafluorochlorobenzene (C_6_F_5_Cl), pentafluorobromobenzene (C_6_F_5_Br), and pentafluoroiodobenzene (C_6_F_5_I) are the reactants reported previously. Theoretically, compared with other reactants, the reactivity of C_6_F_5_Cl may be not ideal, leading to the necessity of the Grignard exchange method. For C_6_F_5_I, the C−I bond has high reactivity, but on the other hand, the reaction involving C_6_F_5_I has poor advantages in consideration of atomic economy. Taken together, C_6_F_5_Br is a relatively appropriate reactant for preparing tetrakis(pentafluorophenyl)borate anions.

For the selection of dehalogenated reagents, butyl lithium reagent and Mg both have a good performance. Butyl lithium reagent, which can even shake C−H bonds, naturally behaves well in dehalogenation reactions. However, LiC_6_F_5_ intermediate is potentially dangerous and may not be conducive to laboratory operation and industrial production. Compared to the former, the intermediate magnesium bromide pentafluorophenyl (C_6_F_5_MgBr) can lay a good foundation for the subsequent boration reaction. 

Boration reagents mainly include BF_3_, BCl_3_, BBr_3_, and B(OCH_3_)_3_. Among them, B(OCH_3_)_3_ can be more favorable to the scaling up of production, on account of its relatively stable properties, further reducing the harsh requirements of the process. 

After boronization, tetrakis(pentafluorophenyl)borate anions can be obtained. If Mg participates in dehalogenation, the magnesium salt (magnesium bromide tetrakis(pentafluorophenyl)borate) generated in the process of industrial scale-up will not be disposed of easily, resulting in low catalyst efficiency. Therefore, researchers further studied the reaction and optimized the process by ion exchange. At present, the reagents utilized in the exchange reaction mainly include potassium reagents, sodium reagents, and ammonium reagents, as exhibited in Table 2.

KF and K_2_CO_3_ have been declared as the potassium reagents for ion exchange [47,48]. Sodium reagents (NaF and Na_2_CO_3_) and ammonium reagents (NH_4_Cl) also show great performance. The ion exchange reaction has solved the residual problem of magnesium salt and has good prospects for industrial production.

It can be summarized from the above statement that most studies on the preparation of ammonium tetrakis(pentafluorophenyl)borate have focused on the synthesis of tetrakis(pentafluorophenyl)borate anions. 

The perfluoroorganometallic reagents react with boron to produce corresponding tetrakis(pentafluorophenyl)borate, which is what these synthesis methods have in common. As a basis for such synthesis methods, each study has been optimized in different aspects to make the operation more concise and safer with more ideal yield and purity. The intense reactivity of *t*-butyl lithium reagents can allow them to react with pentafluorobenzene, which, as a reactant, has more superb atomic economy in comparison with pentafluorohalobenzene. 

However, harsh reaction conditions limit the application and development of butyl lithium reagents. Grignard reagents can be prepared at room temperature and have an ideal yield, which may be a better choice in consideration of the safety and operability of industrial production. In addition, it can effectively help the promotion of industrial production to search for more appropriate boron reagent alternatives.

### 3.2. Preparation for DMAB and MDOAB

With the basis of synthesis of borate anions, the preparation of the DMAB is finally presented in Table 3. *N*,*N*-Dimethylaniline hydrochloride reacts with different intermediates to obtain the final product DMAB with different yields. In comparison with magnesium bromide tetrakis(pentafluorophenyl)borate [44], the reaction in which lithium tetrakis(pentafluorophenyl)borate participated exhibited a good yield [39]. After ion exchange with the magnesium bromide tetrakis(pentafluorophenyl)borate, the yield increased gradually [48,49,50].

For the preparation of MDOAB, the basic preparation method is similar to the above, and the organic cation adopts Me(C_18_H_37_)_2_NH^+^. The preparation method of NMe(C_18_H_37_)_2_ can be found in a previous report [51]. And there is also a report that MDOAB can be obtained from DMAB, which is a novel idea [52]. For the other ammonium tetrakis(pentafluorophenyl)borate, the preparation method is much the same, just with different organic cations used.

## 4. Research Progress of Ammonium Tetrakis(pentafluorophenyl)borate

### 4.1. Types of Ammonium Tetrakis(pentafluorophenyl)borate

According to the preparation above, the ammonium tetrakis(pentafluorophenyl)borate can be mainly divided by two types of cations, including aromatic ammonium and aliphatic ammonium. Typical representatives are dimethylanilinium tetrakis(pentafluorophenyl)borate (DMAB) and *N*-methyl-*N*,*N*-dioctadecylammonium tetrakis(perfluorophenyl)borate (MDOAB).

#### 4.1.1. Dimethylanilinium Tetrakis(pentafluorophenyl)borate

In the olefin polymerization catalyzed by metallocene catalyst, DMAB was used as an organoboride similar to alkyl aluminoxane and alkyl aluminum was used as cocatalyst, showing high polymerization activity [52,53,54]. However, DMABs are insoluble in aliphatic hydrocarbon solvents (such as hexane, cyclohexane, or methylcyclohexane) [52], and olefin polymerization in these solvents is a necessary process for commercialization [55,56,57]. The poor solubility of these cocatalysts can be burdensome in some cases [37,58]. At present, this cocatalyst is mainly utilized in solution polymerization. When the solubility of the cocatalyst is not ideal, the cocatalyst is over-added to promote the dissolution balance so that the cocatalyst in the reaction system can always remain saturated [59]. After activation in a more polar aromatic solvent, the activated complex can be fed into a reactor filled with aliphatic hydrocarbon solvents. However, DMAB is slightly insoluble in toluene, which still requires excess to achieve optimal productivity, so a new class of aliphatic cationic cocatalysts to improve solubility needs to be developed.

#### 4.1.2. N-Methyl-N,N-dioctadecylammonium Tetrakis(perfluorophenyl)borate

More than 20 years ago, Dow introduced a borate with long alkyl chains, *N*-methyl-*N*,*N*-dioctadecylammonium tetrakis(perfluorophenyl)borate ([Me(C_18_H_37_)_2_NH]^+^[B(C_6_F_5_)_4_]^−^, MDOAB) [60]. Due to the long alkyl groups, its solubility in cyclohexane or methyl cyclohexane has been greatly improved. In the catalytic reaction [61,62], the activated complex produced by the MDOAB can be stably dissolved. Wanhua Chemical Group Co., Ltd. (Singapore) [63] proposed a bimetallic catalyst system composed of MDOAB as a cocatalyst, which has excellent thermal stability in the catalytic preparation of olefin polymers and can also produce high-molecular-weight polyolefin. However, MDOABs are highly soluble and cannot be purified by recrystallization, and are easily contaminated with water or Cl-salt impurities that interfere with the activation reaction [64].

### 4.2. The Application of Ammonium Tetrakis(pentafluorophenyl)borate in Polymerization

Organoborates, such as perfluoroarylborates, can interact with the activated metal center to form an active site that can achieve the polymerization [21,27,28,29,32]. The advantage of these organoborate cocatalysts is that they have high catalytic activity and can carry out olefin polymerization under milder reaction conditions. In addition, their catalyst precursors can be obtained through a simple synthetic reaction, which is relatively convenient. These cocatalysts are susceptible to interference from impurities such as moisture and air, which could affect the effect of the catalytic performance. Therefore, although organoborate cocatalysts containing Lewis acidity have certain advantages, their application also faces some challenges, and further research and improvement are needed [34,36,37,65]. The following is a classification summary of their specific applications. 

#### 4.2.1. The Ammonium Tetrakis(pentafluorophenyl)borate as a Substitute for MAO

##### Copolymerization of Ethylene and 1-Octene

In the copolymer of ethylene and α-olefin, when the content of α-olefin in the copolymer is different, it shows unique physicochemical properties. Ethylene and α-olefin copolymers act as physical crosslinks through the crystallization of polyethylene chain segments, thus showing the behavior of thermoplastic elastomers, which has been a major concern of researchers [66,67]. Ethylene/1-octene copolymers are a rapidly developing class of copolymers of ethylene and higher α-olefin.

At present, the ammonium tetrakis(pentafluorophenyl)borate cocatalyst as a substitute for MAO can be applied to the polymerization of ethylene and 1-octene, as shown in Table 4. Lee et al. [68] showed the catalytic performance of group-4 metal complexes with the phosphine-amido ligand derivatives in the copolymerization of ethylene and 1-octene. In the catalytic system, MDOAB is used as the cocatalyst to efficiently achieve copolymerization. Compared with the reaction system with MMAO (7800 kg mol^−1^ h^−1^), it exhibited relatively high copolymerization activity (19,000 kg mol^−1^ h^−1^), indicating that it is a good alternative. In 2015, Lee et al. [69] reported the zirconium and hafnium complexes for the polymerization of ethylene and 1-octene. In the optimization of polymerization conditions, they found that in comparison with MMAO, MDOAB was mixed into the catalytic system and still maintained good catalytic activity (28,000 kg mol^−1^ h^−1^), laying a perfect foundation for follow-up research. Later, Klosin et al. [70] also mentioned the catalytic system including DOMAB to participate in the copolymerization of ethylene and 1-octene, reaching a high catalytic activity of 966,000 kg mol^−1^ h^−1^.

In addition to the academic field, the related research in the industrial field has also attracted considerable attention. Liu et al. [63] disclosed a bimetallic catalyst and the corresponding preparation. In the presence of the catalyst system, high-molecular-weight olefin polymer can be prepared, and the catalyst has excellent thermal stability. In the catalytic system, MDOAB is utilized to efficiently achieve the copolymerization of ethylene and 1-octene. Wanhua Chemical Group Co., Ltd. [71] also uncovered a method for preparing α-olefin copolymer, which includes polymerization of 1-octene and ethylene. The α-olefin is an oligomer prepared from ethylene in the presence of a catalyst system consisting of MDOAB. The invention uses the α-olefin (mainly 1-octene) as raw material to prepare polyolefin elastomer (POE), which has fewer impurities, can obtain POE products with high polymerization activity and ideal 1-octene incorporation, while enhancing the elasticity properties of POE products, which have great application potential.

##### The Polymerization of Ethylene and Styrene Derivatives

Polyethylene and polystyrene are two kinds of plastics known for their wide application [72,73]. The copolymerization of ethylene and styrene has never been overlooked and research on them has also been carried out to produce a completely brand-new property of the material [74]. DMAB can also be used in place of MAO for the polymerization of ethylene and styrene derivatives. Early studies [75] using a combination of MAO and Ti compounds to catalyze ethylene and styrene showed that only a fraction of styrene was transferred to the copolymer. And the transition metal catalysts used in the early stage are highly oxygenophilic and easily poisoned by polar groups; thus, the coordination and insertion of polar monomers such as styrene derivatives can easily cause reaction termination, resulting in a relatively poor insertion rate of polar monomer involved in the reaction. Cui et al. [76] found the catalytic behaviors of the combination of DMAB and rare earth metal-based catalyst for the polymerization of ethylene and styrene derivatives, as seen in Figure 5. This study obtained a perfect alternate insertion rate, indicating that DMAB plays a major role in the copolymerization of this catalytic system. In addition, Cui et al. [77] also achieved the copolymerization of ethylene and para-methoxystyrene with the participation of DMAB as a cocatalyst, which demonstrates that DMAB has a widespread application prospect.

##### The Polymerization of Ethylene and Unsaturated Carboxylate

The catalytic system composed of DMAB can also catalyze the polymerization of olefin and unsaturated carboxylate in place of the MAO. The invention in [78] discloses a novel polymer preparation method. Olefin and unsaturated carboxylate react in the presence of the catalyst composition, which may include DMAB, making the polymerization process more efficient. The polymerization activity of DMAB is comparable to that of MAO. Spherical and/or sphere-like polymers are prepared directly without subsequent processing steps such as granulation, and the polymerization products obtained are not easy to scale in the reactor, making the transportation convenient. This method overcomes the challenges of polymer transportation, solvent removal, and granulation caused by scaling in polymerization equipment; thus, it has a good industrial application prospect. Similarly, SINOPEC (Beijing) Research Institute of Chemical Industry Co., Ltd. (Beijing, China) [79] applied DMAB cocatalyst to the copolymerization of ethylene and unsaturated carboxylate. When the catalyst system including DMAB worked, it showed high polymerization activity. The copolymer containing spherical and/or sphere-like olefin/unsaturated carboxylate can be directly obtained as a result of this method, possessing ideal morphology and having high-profile application prospects. Compared with the process currently used in the industry, saponification is omitted, which can make the preparation simpler.

##### Copolymerization of Styrene and Butadiene

Styrene-butadiene rubber (SBR) copolymers are obtained through using free radical initiators in emulsion polymerization and alkyl lithium reagents in solution polymerization. Although these techniques performed well, controlling polymer tacticity and stereoregularity can be limited. Bowen et al. prepared SBR copolymers that are not contaminated by appreciable amounts of homopolymers with MAO as a cocatalyst [80]. In previous reports, ammonium tetrakis(pentafluorophenyl)borate can be a substitute for MAO in the copolymerization of styrene and butadiene. Shi et al. [81] reported a catalytic system consisting of a rare earth metal complex containing tetrahydrofluorenyl ligands as catalyst and DMAB as cocatalyst for the polymerization of styrene and butadiene. The catalytic system can convert 500 equiv of styrene in a yield of 99% within 15 min, promote the 1,4-regular polymerization of butadiene with highly syndiotactic polystyrene blocks and 1,4-specific PBD blocks.

##### Homopolymerization of 4-methyl-1-pentene

Poly(4-methyl-1-pentene) is a thermoplastic resin developed a long time ago [82]. Due to its low density, superior heat resistance, UV light transmittance, excellent electrical insulation, and chemical resistance [83], it has attracted more and more interest from researchers. Meanwhile, it can be molded by injection molding, blow molding, extrusion, and other methods. The main applications are the manufacture of medical equipment (such as syringes), physical and chemical experimental equipment, electronic cookers, baking trays, stripping paper, heat-resistant wire coating, and other fields [84].

Stephen A. Miller et al. [85] made Me_2_Si(*η*^1^-N-*^t^*Bu)(*η*^1^-C_29_H_36_)ZrCl_2_·OEt_2_ and MAO from a catalytic system, which showed the high activity (5160 kg mol^−1^ h^−1^) and syndioselectivity for the production of poly(4-methyl-1-pentene) with a melting temperature of 215 °C. The DMAB also participates in the homopolymerization of 4-methyl-1-pentene as a cocatalyst. The hot paper by Sita et al. [86] indicated the synthesis and properties of poly(4-methyl-1-pentene). The results showed that poly(4-methyl-1-pentene) with a yield of 3.65 g can be obtained under the catalysis of a system including DMAB, as shown in Figure 6. In 2024, Sita et al. [87] showed that cyclopentadienyl amidinate (CPAM) group-4 metal-active substances with different stereoselectivity are synthesized during the live coordination chain transfer polymerization (LCCTP) of 4-methyl-1-pentene. The control production of poly(4-methyl-1-pentene) was realized with the DMAB as cocatalyst with a yield of 1.99 g, which displayed that the synthesis of polymer materials with adjustable range of elastic properties could be achieved in “one-pot” fashion.

##### Homopolymerization of Isoprene

DMAB was also used in a novel system to catalyze the homopolymerization of isoprene, except that the above polymerizations take the place of MAO. The summary of homopolymerization of isoprene with different cocatalysts is shown in Table 5. In 2002, Masuda et al. [88] reported on Neodymium (III) isopropoxide [Nd(O*i*-Pr)_3_] and MAO as the catalyst system used for isoprene polymerization with high *cis*-1,4 stereoregularity (>ca. 90%) and a perfect number-average molecular weight. And the research about DMAB has gone through a long process.

Later, Destarac et al. [89] adopted Nd-type complex as a catalyst and DMAB as the cocatalyst for homopolymerization of isoprene. The *cis*-selectivity is up to 92% and the activity is relatively poor (68 kg mol^−1^ h^−1^). Visseaus et al. [90] used a catalytic system including the rare earth complex and DMAB to achieve the polymerization of isoprene, which obtained an ideal catalytic activity. However, the *cis*-selectivity was not satisfactory. After that, Masuda et al. [53] also made Nd catalysts and used DMAB to catalyze the polymerization of isoprene, exhibiting superior catalytic activity. Moreover, the *cis*-structural selectivity is ideal (90.4%) and the molecular weight distribution (*M*_w_/*M*_n_ = 2.27) can be relatively narrow.

Wang et al. also developed the well-designed rare earth metal dialkyl complexes and ammonium tetrakis(pentafluorophenyl)borate as the catalytic system for isoprene polymerization with high conversion and high stereoselectivity [91]. Maichle-Mössmer et al. [92] took [Ln(Al*i*-Bu_4_)_2_] (Ln = Sm, Eu, Yb) as catalysts for isoprene polymerization. The [Yb(Al*i*-Bu_4_)_2_]/[HNPhMe_2_][B(C_6_F_5_)_4_] also showed impressive catalytic activity. Another milestone has been achieved in the participation of DMAB in catalyzed isoprene polymerization, making it a satisfactory substitute. 

#### 4.2.2. Ammonium Tetrakis(pentafluorophenyl)borate as an Excellent Performer Compared with MAO

##### Coordinative Chain Transfer Polymerization (Hompolymerization of Ethylene)

In comparison with MAO, ammonium tetrakis(pentafluorophenyl)borate behaves well in specific catalytic reactions. Kempe et al. [93] reported the realization of a highly efficient and controlled polymerization of ethylene through coordinative chain transfer polymerization. The DMAB plays a critical role in Zr catalytic systems, which enable catalytic activity to be 11,500 kg mol^−1^ h^−1^, obviously superior to the catalytic activity for this kind of reaction reported so far with MAO [93,94].

##### Oligomerization of 1-Decene

Hydrocarbons with high boiling points are used as base stocks for engine oils and lubricants. Hydrocarbon-based stocks can consist of four types. The first type is dewaxed and deasphalted crude oil distillate; the second one is generated by catalytic hydrogenation of the first one. The third category is semi-synthetic oil produced by catalytic hydrocracking of high-grade crude oil fractions, accompanied by partial conversion of straight chain alkanes to branchchain saturated hydrocarbons. The fourth class is fully synthetic oils composed of hydrogenated α-olefin oligomers, commonly referred to as polyalpha-olefin (PAO) oils or PAOs [95,96,97]. Hydrogenated oligomers of higher α-olefin (mainly, 1-decene) are the base stocks for transmission fluids, lubricants, PAOs, and greases. Zirconium catalyst compositions have been proved to be highly competitive in α-olefin oligomerization and have been widely implemented for industrial fields. DMAB as the cocatalyst represents a great advantage compared with MAO. Pavel V. Ivchenk et al. [98] reported that 10 eq. of MAO as the cocatalyst is effective in α-olefin oligomerization including 1-decene, and moderate yields (~40%) of trimer-pentamer fractions were achieved. Later, the same research team [99] tried the heterocene-catalyzed method for oligomerization of 1-decene. The MMAO and DMAB are both the activators for the oligomerization. They achieved at least 50% yields of lightweight oligomers (degree of polymerization (*DP*_n_) = 3–4) with MMAO (10 eq.) or by DMAB (1.5 eq.), as shown in Figure 7. And the conversion rate of the catalytic system with DMAB is higher than that with MMAO. The oligomer obtained by the research is a kind of high-quality PAO base stock with viscosity properties that far exceed those of PAOs in the industrial field.

#### 4.2.3. Unique Role of Ammonium Tetrakis(pentafluorophenyl)borate as the Cocatalyst 

##### The Production of Telechelic Polyolefins

Ammonium tetrakis(pentafluorophenyl)borate plays a unique role as the cocatalyst in some specific polymerization reactions. Sita et al. [100] reported a general synthesis strategy for the production of phenyl-group-terminated poly(α-olefins) in the presence of DMAB in the catalytic composition, as shown in Figure 8. They concluded that phenyl groups, as the synthons of other telechelic polyolefin functional groups, can be “uncovered” by simple high-yield post-polymerization reactions. They demonstrated that polyolefin-based block copolymers can be easily derived from a wide range of telechelic polyolefins. The synthesis of polyolefin-polyester diblock copolymers and triblock copolymers has demonstrated the value of these materials as building blocks for structural classes of polyolefin-based synthetic polymers.

##### The Preparation of Silane-End-Functionalized Polymers

Silane-end-functionalized polymers containing different organofunctional silane groups, such as carbosilane, hydrosilane, chlorosilane, alkoxy-silane, and acetoxy-silane have important applications in adhesives, coatings, sealants, and elastomers [101,102]. And it can also be used as a precursor for conversion into telechelic polymers and other macromolecular structures. Currently, DMAB was also involved in the preparation of silane-end-functionalized polymers in the key role. Li et al. [103] have developed a functional approach for synthesizing the silane-end-functionalized linear and star polyaryl-isocyanates through the full combination of hydrosilylation reaction and ionic polymerization. The borate composition is mainly used to catalyze the cationic polymerization of aryl isocyanates. And the configuration, end group type, and molecular weight of these silan-end-functionalized polyaryl-isocyanates can be easily adjusted by hydrogenation of hydrosilanes. DMAB, as one of several cocatalysts, has also been tested in this reaction, indicating its wide application potential and laying a great foundation for subsequent research.

##### Some Specific Cyclopolymerizations Associated with 1,6-Heptadiene

In recent years, DMAB and transition metal complexes forming catalytic systems can be tried and applied in some further specific reactions. In 2013, Kaitlyn E. Crawford and Lawrence R. Sita [104] found that DMAB and transition metal (Zr and Hf) complexes can participate in the coordination polymerization of 1,6-heptadiene and different physical forms of poly(1,3-methylene cyclohexane) (PMCH), as presented in Figure 9. This study lays the foundation for DMAB to participate in some specific polymerization reactions about 1,6-heptadiene.

Later, Kaitlyn E. Crawford and Lawrence R. Sita [105] reported another cooperation between DMAB and the transition metal complexes for synthesizing poly(1,3-methylenecyclohexane)-b-atactic polypropene-b-poly(1,3-methylenecyclohexane) (PMCH-b-aPP-b-PMCH) polyolefin triblock copolymer, as shown in Figure 10. The hardness and softness of different blocks can vary. By changing the mass fraction of different blocks, unique-peculiarity polyolefin block copolymers can be obtained. The controllable “precision” polyolefin can help satisfy the technical needs of the future society.

In 2023, Sita et al. [106] reported that a systematic investigation about the synthesis and characterization a series of amorphous atactic *cis*, *trans* poly(methylene-1,3-cyclopentane-stat-cyclohexane) statistical copolymers was carried out innovatively. In this study, the copolymerization of 1,5-hexadiene and 1,6-heptadiene with different initial feed ratios was attempted, in order to produce different grades of products under the catalysis of DMAB and Hf complexes. The shape of the product can vary with the proportion of five-membered and six-membered cycloalkane repeating units. As the content of the six-membered ring increases, the glass transition temperature (*T*g) of the product can be increased from −16 °C to 100 °C, although not in a strictly linear manner, as seen in Figure 11. It was further shown that a small level of six-membered ring content is sufficient to destroy the crystallization of limiting atactic *cis*, *trans* poly(methylene-1,3-cyclopentane) (PMCP) homopolymers with a melting temperature of 98 °C, which lays the foundation for potential future technical applications of this unique class of copolymers.

DMAB and MDOAB as cocatalysts can be implemented in various aspects of polymerization. In terms of economic cost, they are the ideal substitutes for MAO, and some are even better than MAO in some specific reactions. Compared with MDOAB, DMAB has been studied more thoroughly and applied to a wider range of cases. And it is a highly competitive cocatalyst worthy of further study. However, in combination with certain catalysts, DMAB has not been able to replace the effects of MAO yet. For example, Nomura et al. [107] indicated that cyclopentadienyl ketamide titanium (IV) complex−MAO catalyst system showed high catalytic activity (519 kg mol^−1^ h^−1^) in the copolymerization of ethylene, while DMAB has relatively poor activity (79.2 kg mol^−1^ h^−1^). And Pellecchia et al. [108] found the catalytic behavior of zirconium dialkyl pre-catalysts with pyrrolyl- or indolyl-pyridyl-amido ligands. The results show that under specific experimental conditions, the catalytic activity of employing DMAO as the cocatalyst (1.4 × 10^6^ g mol^−1^ h^−1^) is much higher than that of DMAB (trace), indicating that there is still plenty of room for research and improvement of DMAB.

## 5. Conclusions

Currently, ammonium tetrakis(pentafluorophenyl)borate as cocatalyst is critical in catalytic systems and can be of wide interest for the application of polymerization. In comparison with MAO, ammonium tetrakis(pentafluorophenyl)borate is relatively inexpensive, and in addition, it is chemically stable and not easy to react with moisture and air, which further reduces the storage and transportation costs of the cocatalyst. As the ideal alternative to MAO, it is considered to play a unique role in the production of telechelic polyolefins, silane-end-functionalized polymers, and cyclopolymerizations associated with 1,6-heptadiene. To date, ammonium tetrakis(pentafluorophenyl)borate is still not a complete replacement for MAO in specific polymerization fields, which deserves further in-depth research. When evaluating a metallocene catalyst, its catalytic application range, stability, molecular weight of the polymer product, molecular weight distribution of the polymer product, and many other factors need to be investigated as much as the catalytic activity. Therefore, it is of great value to modify the structure of metallocene compounds and ammonium tetrakis(pentafluorophenyl)borate cocatalysts with persistence in order to realize the characteristic catalytic performance. For modified ammonium tetrakis(pentafluorophenyl)borate, changing aromatic ring types of *N*,*N*-dimethylaniline or introducing substituents into aromatic rings may be feasible research avenues. It is also viable to increase the solubility of ammonium tetrakis(pentafluorophenyl)borate in alkane by turning methyl into long-chain alkyls. However, it is important to emphasize that as a typical electron donor group, long-chain alkyls can enhance the alkalinity of aromatic amines, which may hinder the dealkylation of transition metal to a certain extent. And it should be noted that overly complex modifications will increase costs, significantly affecting the advancement of industrialization. The preparation process of ammonium tetrakis(pentafluorophenyl)borate is also worthy of further study. In terms of the industrial cost and safety, it has significant application value to find an alternative to organometal reagent for preparing the borate in order to solve the problem of sensitivity to moisture and air in the preparation process.

## Data Availability

Not applicable.

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
