# Peer review of "Ammonium Tetrakis(pentafluorophenyl)borate: Preparation and Application in Olefin Coordination Polymerization as the Cocatalyst Compound"

_polymers, 2024, doi:10.3390/polym16121689_

Round 1

Reviewer 1 Report

Comments and Suggestions for Authors

General comments

The paper deals with a new generation of co-catalysts for metallocene olefin polymerization, Ammonium Tetrakis(pentafluorophenyl)borate co-catalysts. It focuses on the preparation methods, the differences compared with MAO, first applied as an efficient co-catalyst in the field of olefin polymerization, and the research progress in this field.

The introduction is well organized, complete in references.

The authors correctly illustrate the role of catalysts and co-catalysts (Lewis acids) in the olefin polymerization process, the critical issues in the use of MAO, and the development of this new class of co-catalysts, the perfluoroaryl borates. These last represent a valid alternative to MAO, thanks to the synergy between the electronic and steric effect of perfluorophenyl group, which significantly improves their catalytic performance.

The authors explain well the state of the art on these co-catalysts, and, through a correct subdivision and organization of the sections, they well illustrate the preparation, the research progress and the applications in various polymerization processes in comparison with MAO.

However, for the review to be considered for publication, the authors must adequately address the issues raised by the reviewer which are set out below.

Detailed comments

1)     In the introduction, the authors state: “The activation mechanism of perfluoroaryl borate and MAO as a cocatalyst was illustrated in Scheme 1. The transition metal complex was firstly alkylated by alkyl aluminum and then an alkyl group was abstracted by the borate to form the transition metal cation. These cations form a loose ion pair with the borate and then reacts with the monomer, realizing the chain initiation and propagation. “

I can't find a good correspondence between the text and what it is reported in Scheme 1. The authors should clearly distinguish what happens in mechanism (a) with MAO and in mechanism (b) with perfluoroaryl borate.

2)     The description of the caption for "Scheme 1" and for all the schemes of the review must be reported.

3)     In Scheme 3, the MDOAB compound is illustrated and not DMAB one, as reported in the text. In the same way, in the Scheme 4, the DMAB compound is shown and not the MDOAB one, as reported in the text. Please correct.

4)     The structural formulas of MDOAB and DMAB should be improved: the H atom should be bonded to nitrogen atom and the geometries of the cations could be better represented.

5)     The title of the subparagraph 4.2.1.6 is the same as that of 4.2.1.5. Please correct.

6)     In the subparagraph 4.2.2.2 oligomerization of 1-decene. Please correct the first word with a capital letter.

Comments on the Quality of English Language

Minor editing of English language required.

Author Response

Manuscript ID: polymers-3023242

Type of manuscript: Review

Title: Ammonium Tetrakis(pentafluorophenyl)borate: Preparation and Application in Olefin Coordination Polymerization as the Cocatalyst Compound

Dear Professor,

Thank you very much for your important and constructive comments on our manuscript. We have learned a lot from your comments that we may have previously overlooked. We have carefully studied your comments and have made careful changes to the manuscript based on your comments. The following will be a point-by-point response to your comments. All revisions are highlighted in the manuscript. We hope that our answers will satisfy you.

Best Regards

Wei Wang

Reviewer #1

The paper deals with a new generation of co-catalysts for metallocene olefin polymerization, Ammonium Tetrakis(pentafluorophenyl)borate co-catalysts. It focuses on the preparation methods, the differences compared with MAO, first applied as an efficient co-catalyst in the field of olefin polymerization, and the research progress in this field. The introduction is well organized, complete in references. The authors correctly illustrate the role of catalysts and co-catalysts (Lewis acids) in the olefin polymerization process, the critical issues in the use of MAO, and the development of this new class of co-catalysts, the perfluoroaryl borates. These last represent a valid alternative to MAO, thanks to the synergy between the electronic and steric effect of perfluorophenyl group, which significantly improves their catalytic performance.The authors explain well the state of the art on these co-catalysts, and, through a correct subdivision and organization of the sections, they well illustrate the preparation, the research progress and the applications in various polymerization processes in comparison with MAO. However, for the review to be considered for publication, the authors must adequately address the issues raised by the reviewer which are set out below.

Point 1. In the introduction, the authors state: “The activation mechanism of perfluoroaryl borate and MAO as a cocatalyst was illustrated in Scheme 1. The transition metal complex was firstly alkylated by alkyl aluminum and then an alkyl group was abstracted by the borate to form the transition metal cation. These cations form a loose ion pair with the borate and then reacts with the monomer, realizing the chain initiation and propagation.” I can't find a good correspondence between the text and what it is reported in Scheme 1. The authors should clearly distinguish what happens in mechanism (a) with MAO and in mechanism (b) with perfluoroaryl borate.

Response: The reviewer's suggestion is very reasonable. We have revised Scheme 1 and the corresponding text to distinguish the activation mechanism of MAO and perfluoroaryl borate as the cocatalyst clearly on page 2, highlighted in manuscript.

Point 2. The description of the caption for "Scheme 1" and for all the schemes of the review must be reported.

Response: The reviewer's suggestion is very reasonable. We have added the description of the caption for all the schemes of the review.

Point 3. In Scheme 3, the MDOAB compound is illustrated and not DMAB one, as reported in the text. In the same way, in the Scheme 4, the DMAB compound is shown and not the MDOAB one, as reported in the text. Please correct.

Response: The reviewer's suggestion is very reasonable and we do apologize for our careless mistakes. We have corrected the corresponding content.

Point 4. The structural formulas of MDOAB and DMAB should be improved: the H atom should be bonded to nitrogen atom and the geometries of the cations could be better represented.

Authors response: The reviewer's suggestion is very reasonable. We have revised the structural formulas of DMAB and MDOAB in Scheme 3 and Scheme 4.

Point 5. The title of the subparagraph 4.2.1.6 is the same as that of 4.2.1.5. Please correct.

Response: The reviewer's suggestion is very reasonable and we do apologize for our careless mistakes. We have corrected the corresponding text.

Point 6. In the subparagraph 4.2.2.2 oligomerization of 1-decene. Please correct the first word with a capital letter.

Response: The reviewer's suggestion is very reasonable and we do apologize for our careless mistakes. We have corrected the corresponding text.

Reviewer 2 Report

Comments and Suggestions for Authors

This review article provides a crucial overview of the current state and potential of cocatalysts in metallocene catalysis for olefin polymerization.

It focuses on ammonium tetrakis(pentafluorophenyl)borate as a promising alternative to MAO, highlighting its cost-effectiveness and efficiency in polymerization. Overall, the manuscript is well-written. The author has critically analyzed prior work and has summarized it well into a comprehensive review article.

The review includes approximately 35% of references published in the last 10 years, which is commendable.

No language, spelling, or grammatical mistakes were found in the manuscript.

However, my only concern is with the introduction section. It is repetitive and overly detailed, which can obscure the main points. Additionally, the transition from discussing MAO to perfluoroaryl borates lacks clarity and consistency, making it difficult for the reader to follow the progression of the article. I suggest the author rework the introduction section based on my observations.

I recommend the publication of the manuscript once the author addresses the above concern.

Author Response

Manuscript ID: polymers-3023242

Type of manuscript: Review

Title: Ammonium Tetrakis(pentafluorophenyl)borate: Preparation and Application in Olefin Coordination Polymerization as the Cocatalyst Compound

Dear Professor,

Thank you very much for your important and constructive comments on our manuscript. We have learned a lot from your comments that we may have previously overlooked. We have carefully studied your comments and have made careful changes to the manuscript based on your comments. The following will be a point-by-point response to your comments. All revisions are highlighted in the manuscript. We hope that our answers will satisfy you.

Best Regards

Wei Wang

Reviewer #2

This review article provides a crucial overview of the current state and potential of cocatalysts in metallocene catalysis for olefin polymerization. It focuses on ammonium tetrakis(pentafluorophenyl)borate as a promising alternative to MAO, highlighting its cost-effectiveness and efficiency in polymerization. Overall, the manuscript is well-written. The author has critically analyzed prior work and has summarized it well into a comprehensive review article. The review includes approximately 35% of references published in the last 10 years, which is commendable. No language, spelling, or grammatical mistakes were found in the manuscript. 

Point 1. However, my only concern is with the introduction section. It is repetitive and overly detailed, which can obscure the main points. Additionally, the transition from discussing MAO to perfluoroaryl borates lacks clarity and consistency, making it difficult for the reader to follow the progression of the article. I suggest the author rework the introduction section based on my observations. I recommend the publication of the manuscript once the author addresses the above concern.

Response: Thanks for reviewer's reasonable suggestions. We have reworked the introduction section for making the main points stand out and the paper logical. The revised parts are highlighted in the manuscript.